# Depth-Breadth Synergy in RLVR: Unlocking LLM Reasoning Gains with Adaptive Exploration

## Abstract

Reinforcement Learning with Verifiable Reward (RLVR) is a powerful method for enhancing the reasoning abilities of Large Language Models, but its full potential is limited by a lack of exploration in two key areas: **Depth** (the difficulty of problems) and **Breadth** (the number of training instances). Our analysis of the popular GRPO algorithm reveals a bias that down-weights difficult, low-accuracy problems, which are crucial for improving reasoning skills. To address this, we introduce Difficulty Adaptive Rollout Sampling (DARS), a method that re-weights difficult problems by using targeted, multi-stage rollouts. This approach increases the number of rollout outcomes for these harder problems according to our proposed re-balancing schedules and leads to consistent gains in *Pass@K*. We also found that simply enlarging the rollout size isn't effective and can even harm performance. We also investigated the role of breadth by scaling the batch size and using full-batch updates. This significantly improved *Pass@1* performance by maintaining high token-level entropy, which indicates continued exploration and reduced gradient noise. Finally, we present DARS-Breadth, a combined approach that uses DARS with a large breadth of training data. This method demonstrates simultaneous gains in both *Pass@K* and *Pass@1*, confirming that depth (adaptive exploration) and breadth (scaling the training data) are orthogonal and essential dimensions for unlocking the full reasoning power of RLVR.

## 1 Introduction

The emergence of reasoning-centric Large Language Models (LLMs) exemplified by OpenAI-o1 (Jaech et al., 2024), DeepSeek-R1 (Guo et al., 2025), and Kimi-1.5 (Team et al., 2025), has pushed the frontier of LLM capability, especially for demanding tasks in complex reasoning such as mathematics and programming. Unlike conventional instruction tuning that relies on human-labeled data or RLHF pipelines that demand an auxiliary, well-trained reward model (Ouyang et al., 2022a; Achiam et al., 2023; Grattafiori et al., 2024), this leap is driven by large-scale Reinforcement Learning with Verifiable Rewards (RLVR; Guo et al. 2025; Zeng et al. 2025) for which correctness can be automatically and deterministically checked. The rewards of RLVR are granted solely when a model's output matches the ground-truth mathematical answer or passes all unit tests for code, allowing scalable verification without manual labeling. RLVR is now regarded as a promising path toward self-evolving LLMs, potentially bringing us closer to more powerful intelligence.

However, existing RLVR frameworks inadequately address the interplay between exploration depth (difficulty scaling) and breadth (iteration instance quantity scaling), which leads to insufficient performance gain for both *Pass@1* and *Pass@K*. In this paper, we conduct a systematic analysis of two under-exploited dimensions in RLVR: Depth and Breadth.

For the dimension of **depth**, our investigation reveals that existing methods of GRPO (Shao et al., 2024) and its variants (Yu et al., 2025; Liu et al., 2025b), while adept at estimating the advantage of a single rollout, are undermined by a distorted cumulative advantage at the group level. This distortion disproportionately allocates attention to instances of medium difficulty, neglecting high-difficulty instances indispensable for complex reasoning, as illustrated in Figure 2. This bias fundamentally limits depth, the hardest problems a model can learn to solve, and constrains *Pass@K* performance. To counteract this depth neglect, we propose Difficulty-Adaptive Rollout Sampling

(**DARS**). DARS performs a lightweight first-stage rollout to estimate per-problem accuracies, then allocates additional compute via targeted multi-stage rollouts to low-accuracy problems. By expanding sampling on hard problems, DARS re-weights the cumulative advantage, making it easier for LLMs to learn 'deep' samples and improving *Pass@K* performance.

We further identify **breadth** as the instance quantity consumed in a single iteration. We observe that breadth has a significant impact on the LLM's performance and continuous exploration capability, as shown in Figure 4. We significantly increase the training batch size and replace PPO-minibatch updates with full-batch updates for multiple PPO-epochs. This seemingly simple change dramatically improves *Pass@1* and sustains high token-level entropy throughout training, suggesting that breadth acts as implicit entropy regularization that delays premature convergence. Importantly, the gains from breadth are complementary to those from depth: we present **DARS-Breadth** that combines our DARS with large-breadth training, producing simultaneous boosts in both *Pass@K* and *Pass@1*. Our contributions can be summarized as follows:

- We conduct a systematic analysis on depth and breadth in RLVR, and uncover the depth bias in GRPO: cumulative advantage silently underweights low-accuracy, high-difficulty samples, capping *Pass@K* performance.
- We introduce DARS, which reallocates compute from medium difficulty problems to the hardest problems via multi-stage rollout sampling. DARS re-weights the cumulative advantage distribution and quantitatively expands the sparse reward signals for difficult problems. In practice, DARS significantly improves *Pass@K* over multiple benchmarks.
- We further illustrate that large breadth in RLVR training matters for the *Pass@1* performance. Moreover, by combining DARS with large breadth training, we reveal the complementarity of Depth and Breadth in RLVR and acquire simultaneous boosts in both *Pass@K* and *Pass@1* performance.

## 2 UNDERSTANDING RLVR FROM DEPTH AND BREADTH

### 2.1 DEPTH: THE HARDEST PROBLEM SAMPLED IN RLVR

We first identify **Depth** as the hardest problem that can be correctly answered in the RLVR training process. In the GRPO training process, groups whose entire rollouts yield incorrect answers suffer from gradient vanishing. Hence, sampling high-difficulty questions with correct reasoning paths is crucial for LLM training. We first show that merely increasing rollout size does not consistently yield significant gains in *Pass@K* performance, and sometimes can even be harmful. We then quantify GRPO's cumulative advantage and highlight its under-weighting of high-difficulty samples.

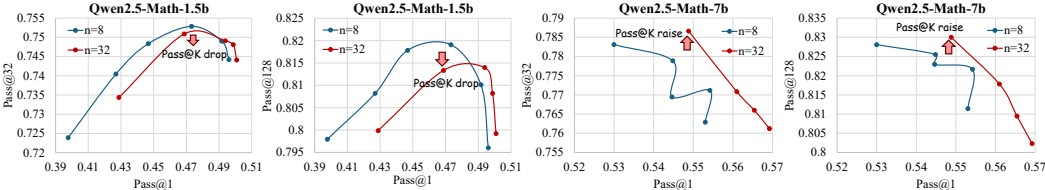

Figure 1: Training dynamics of *Pass@1* and *Pass@K* performance of Qwen2.5-Math-1.5b and Qwen2.5-Math-7b with different rollout size.

**Naive Scaling of Rollout Size Benefits *Pass@1*, But Not Necessarily *Pass@K*.** We present the training dynamics of *Pass@1* and *Pass@K* performance during the RLVR training process in Figure 1. Enlarging the rollout size allows the sampling of correct solutions to hard problems during training. We originally assumed this would benefit *Pass@K* performance; however, experimental results show that this is not always the case. We find that Qwen2.5-Math-7b can significantly benefit from an increased rollout size, whereas for Qwen2.5-Math-1.5b, naively scaling rollout size can even harm *Pass@K* performance.

**Cumulative Advantage Bias in GRPO Variants hinders the improvement of *Pass@K*.** In the GRPO framework, the advantage estimation is derived by normalizing binary rewards:

$$\hat{A}_i^{std} = \frac{r_i - u}{\sigma}, \quad \hat{A}_i^{nostd} = r_i - u, \tag{1}$$

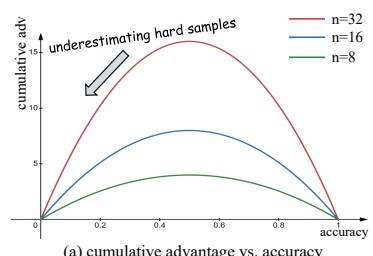 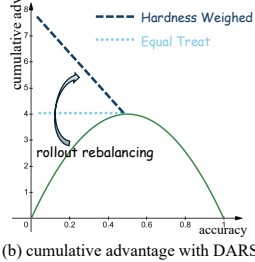

(a) cumulative advantage vs. accuracy      (b) cumulative advantage with DARS

Figure 2: Statistical results of cumulative advantage. Group relative advantage calculation methods underestimate high-difficulty problems. $n$ denotes group size.

where $r_i$ is the binary reward of $i_{th}$ rollout, $u$ is the mean value of the group rewards $u = \text{mean}(\{R_i\}_{i=1}^{G})$ and $\sigma$ is the standard deviation of the group rewards $\sigma = \text{std}(\{R_i\}_{i=1}^{G})$. In the case of binary rewards, $u$ also represents the accuracy of LLM rollouts. Dr. GRPO (Liu et al., 2025b) removes the standard-deviation term from the advantage computation to eliminate question-level difficulty bias, and demonstrates its superiority through extensive experiments. Consequently, the experiments reported in this study were conducted primarily though the Dr. GRPO methodology. We show more results of std-based advantage in Appendix F.1.

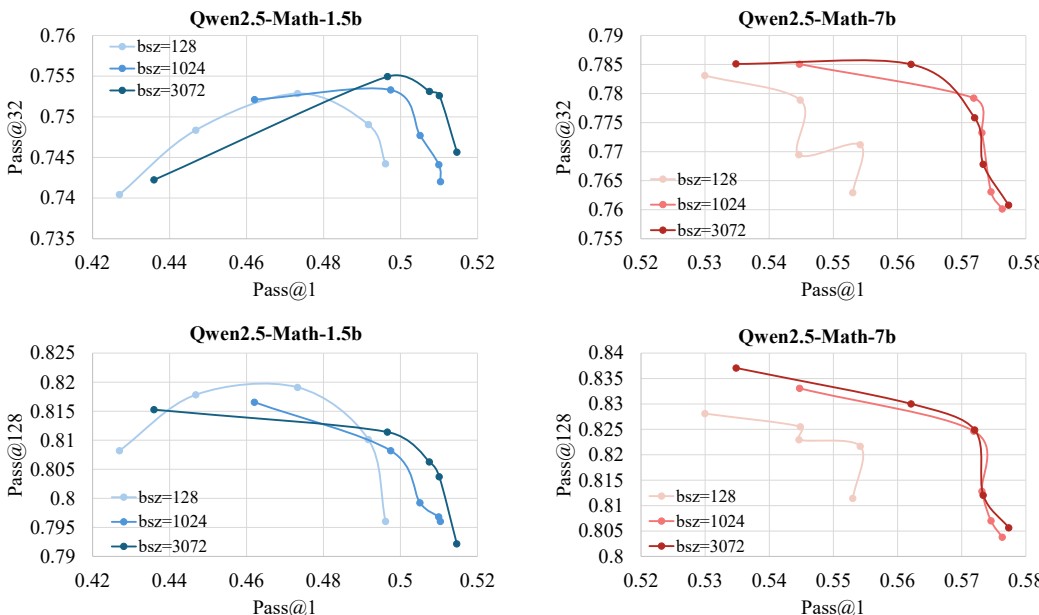

Figure 3: Training dynamics of *Pass@1* and *Pass@K* performance of Qwen2.5-Math-1.5b and Qwen2.5-Math-7b with different batch size.

For a group $G$ with rollout size $N$, we define the **cumulative advantage** of a group as the sum of the absolute values of sample advantages: $\mathcal{A}_{\text{group}} = \sum_{i=1}^{G} |\hat{A}_i|$. The cumulative advantage reflects how much the algorithm weights each sample. Specifically, for Dr. GRPO,

$$\mathcal{A}_{\text{group}} = 2Nu(1-u), \tag{2}$$

The cumulative advantage functional curve is plotted in Figure 2. As shown in the figure, group-based advantage computation funnels its weight toward problems of medium difficulty while largely overlooking those that are highly difficult. This bias limits the *Pass@K* performance of RLVR.

## 2.2 BREADTH: ITERATION INSTANCE QUANTITY IN RLVR

We define **Breadth** as the number of instances used per iteration of the RLVR process. We'll show how increasing the batch size for the RLVR process improves the *Pass@1* performance.

**Breadth Matters for *Pass@1* Performance.** Most studies (Liu et al., 2025b;a; Yan et al., 2025; Fu et al., 2025) conventionally set the batch size to 128. In this subsection, we drastically increase the batch size to 3072 and plot the training dynamics of *Pass@1* and *Pass@32* performance in Figure 3. Naively increasing the batch size brings a *Pass@1* improvement for all models, yet it harms the *Pass@128* performance of Qwen2.5-Math-1.5b. We consider that increasing the quantity of instances used in each iteration makes the gradient direction more accurate and reduces the impact of noise, thereby improving *Pass@1* performance.

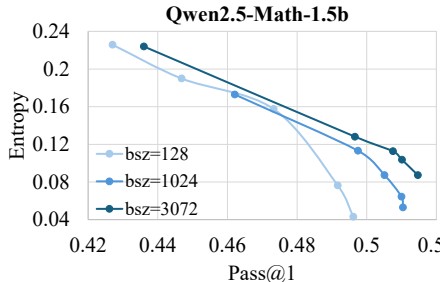 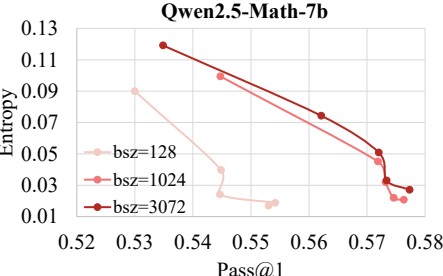

Figure 4: Training dynamics of *Pass@1* performance and token entropy for Qwen2.5-Math-1.5b and Qwen2.5-Math-7b.

**Breadth Sustains Entropy for Model Exploration.**

High token entropy in LLMs indicates strong exploration capabilities. Our analysis shows a relationship between *Pass@1* and token entropy during training. As illustrated in Figure 4, increased training breadth enables LLMs to achieve higher entropy at a given Pass@1 accuracy. We believe a large training breadth acts as a form of entropy regularization, preventing premature convergence and boosting *Pass@1* performance while maintaining high entropy.

## 3 METHODOLOGY

In Section 2, we analyze the bias inherent in group-based advantage computation. To solve this issue, we introduce Difficulty Adaptive Rollout Sampling (**DARS**), which rebalances the cumulative advantage via multi-stage sampling. By further synergizing the depth and breadth training dimensions, we propose DARS-B, which improves both *Pass@1* and *Pass@K*.

### 3.1 DIFFICULTY ADAPTIVE ROLLOUT SAMPLING (DARS)

As shown in Figure 5, given a data batch $\mathcal{B} = \{q_j\}_{j=1}^{M}$ of reasoning questions, DARS operates in two phases: (i) **pre-rollout difficulty estimation** that assigns to each question $q_j$ a scalar difficulty score $x_j \in [0, 1]$; and (ii) **multi-stage rollout re-balancing** that dynamically decides how many additional trajectories $\Delta n_j$ shall be allocated to $q_j$ so that the cumulative advantage for low-accuracy problems is up-weighted. To simplify the subsequent formula representation, we define

$$\mathcal{S}(\hat{a}_j) = 2\hat{a}_j(1 - \hat{a}_j). \tag{3}$$

**Phase 1: Pre-Rollout Difficulty Estimation.** For every $q_j$, we draw a light first-stage rollout consisting of $N^{pre}$ independent trajectories $\{\tau_j^{(i)}\}$. Let the per-trajectory reward be binary, $r_j^{(i)} \in \{0, 1\}$. We define the empirical accuracy

$$\hat{a}_j = \frac{1}{N^{pre}} \sum_{i=1}^{N^{pre}} r_j^{(i)}. \tag{4}$$

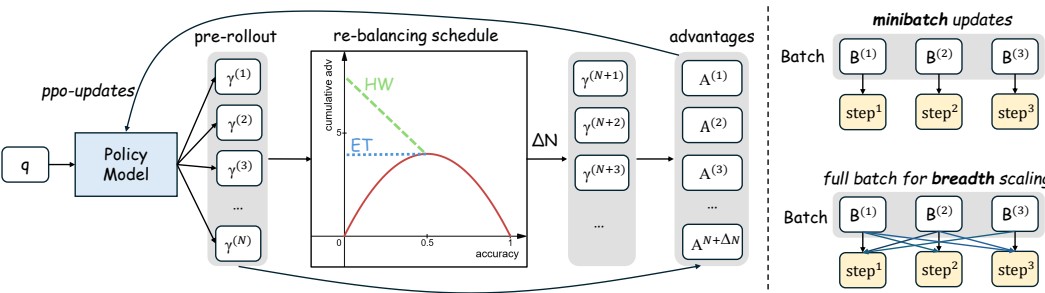

Figure 5: The overall training framework of our Difficulty Adaptive Rollout Sampling (**DARS**) with breadth scaling. Our DARS consists of 2 phases: 1) a pre-rollout stage to evaluate the difficulty of the given question, and 2) a re-balancing rollout stage to adjust the cumulative advantage. For breadth scaling, we replace ppo minibatch as full batch with multiple ppo epochs.

The difficulty score is then set to the complementary accuracy $x_j = 1 - \hat{a}_j$, so that $x_j \approx 1$ for the hardest problems and $x_j \approx 0$ for the easiest ones.

**Phase 2: Multi-Stage Rollout Re-Balancing.** Let $\mathcal{A}_{\text{group}}^N(u)$ denote the cumulative advantage under GRPO for a group whose average accuracy is $u$ with rollout size $N$. We aim to reallocate $\Delta N$ additional trajectories across the mini-batch so that the **effective** cumulative advantage for each question becomes an increasing function of its difficulty. To control the computing cost, we cap the rollout sampling upper limit at $N^{\max}$. To this end, we design two rebalancing schedules.

**Schedule 1: Equal-Treatment (ET).** For every question $q_j$ we enforce the rebalanced cumulative advantage as:

$$\mathcal{A}_{\text{group}}^{ET}(q_j) = \mathcal{A}_{\text{group}}^{N^{pre}}(0.5). \tag{5}$$

We raise the cumulative advantage of all difficulty problems ($\hat{a}_j < 0.5$) to the level achieved by a medium-difficulty problem with accuracy $\hat{a}_j = 0.5$. The required extra trajectories are

$$\Delta n_j^{\text{ET}} = \min\left( \left\lceil \frac{\mathcal{A}_{\text{group}}^{N^{pre}}(0.5) - \mathcal{A}_{\text{group}}^{N^{pre}}(\hat{a}_j)}{\mathcal{S}(\hat{a}_j)} \right\rceil, N^{\max} - N^{pre} \right). \tag{6}$$

**Schedule 2: Hardness-Weighted (HW).** We now impose a monotonically increasing re-weighting that allocates more rollouts to lower-accuracy problems:

$$\mathcal{A}_{\text{group}}^{HW}(q_j) = 2(1 - x_j)\mathcal{A}_{\text{group}}^{N^{pre}}(0.5). \tag{7}$$

This yields

$$\Delta n_j^{\text{HW}} = \min\left( \left\lceil \frac{2x_j \cdot \mathcal{A}_{\text{group}}^{N^{pre}}(0.5) - \mathcal{A}_{\text{group}}^{N^{pre}}(\hat{a}_j)}{\mathcal{S}(\hat{a}_j)} \right\rceil, N^{\max} - N^{pre} \right). \tag{8}$$

## 3.2 DEPTH SYNERGY WITH BREADTH SCALING

Our analysis in Section 2.2 empirically confirms the substantial *Pass@1* improvements from large-breadth training. While DARS primarily optimizes training depth via multi-stage rollout rebalance, its dynamic batch-size adjustments preclude standard PPO-style mini-batch updates. To resolve this architectural constraint while leveraging breadth benefits, we replace PPO's mini-batch updates with full-batch gradient descent across multiple PPO epochs, as illustrated in Figure 5. This modification ensures compatibility with DARS's dynamic allocation while maximizing effective training breadth per optimization step. We term this integrated approach **DARS-Breadth**, unifying depth-adaptive sampling with breadth maximization.

Full-batch training offers two key advantages: (1) elimination of mini-batch gradient noise, and (2) sustained token-level exploration, acting as implicit regularization against premature convergence. The resulting framework demonstrates complementary gains—DARS improves *Pass@K* through depth optimization, while large-breadth training enhances *Pass@1*, highlighting their synergistic roles in RLVR optimization.

## 3.3 TRAINING TARGET

We adopt the clipped objective of GRPO without the KL penalty term. Following Dr. GRPO, we likewise remove the response length handling from the GRPO target. Specifically, for a problem $q$ sampled in training data $\mathcal{D}$, the training target is formalized as:

$$\mathcal{J}(\theta) = \mathbb{E}_{(q \sim \mathcal{D}, \{o_i\}_{i=1}^{\mathcal{G}} \sim \pi_{\theta_{old}}(q))}$$

$$\left[ \frac{1}{G} \sum_{i=1}^{G} \sum_{t=1}^{|o_i|} \left( \min \left( r_{i,t}(\theta) \hat{A}_{i,t}, \ \mathrm{clip}\left( r_{i,t}(\theta), 1 - \varepsilon, 1 + \varepsilon \right) \hat{A}_{i,t} \right) \right) \right], \quad (9)$$

where

$$r_{i,t}(\theta) = \frac{\pi_\theta(o_{i,t} \mid q, o_{i,<t})}{\pi_{\theta_{old}}(o_{i,t} \mid q, o_{i,<t})}. \quad (10)$$

The token advantage $\hat{A}_{i,t}$ is computed using Equation 1.

## 4 EXPERIMENTS

### 4.1 SETUP

**Evaluation and Training:** We evaluate the RLVR process using 5 widely used mathematical reasoning benchmarks: MATH-500 (Lightman et al., 2023), OlympiadBench (He et al., 2024), MinervaMath (Lewkowycz et al., 2022), AIME24, and AMC23. We combine all of the evaluation benchmarks to report *Pass@1 (Avg@128)* and *Pass@K* performance. The training data used in this work is OpenR1-45K, which is a subset of OpenR1-Math-220k (Hugging Face, 2025). More details are shown in Appendix E.

**Baseline and Methods:** We compare with: (1) *RLVR-baseline*: Dr. GRPO with rollout size 8 and batch size 128. (2) *Depth-Naive*: Simply increasing the rollout size to 32. (3) *Breadth-Naive*: Simply increasing the batch size to 3072. (4) *DARS-ET/HW*: Our algorithm introduced in Section 3.1 with Equal-Treat/Hardness-Weighted schedule, using batch size 128 and $N^{\max} = 32$. (5) *DARS-ET/HW-Breadth*: Our Depth-and-Breadth synergy algorithm introduced in Section 3.2, using batch size 3072 and $N^{\max} = 32$. For all methods, the number of PPO mini-steps is uniformly set to 2.

**Evaluation Protocol:** For all baselines, we select the checkpoint with the best *Pass@1* performance for reporting. For DARS, we selected the checkpoint that achieved the best *Pass@128* performance after surpassing the baseline *Pass@1* performance. Table 1 summarizes the *Avg@128* performance on each benchmark, the overall *Pass@1* across all test data, and the *Pass@128* performance.

### 4.2 MAIN RESULTS

Breadth scaling delivers a clear and consistent boost to Pass@1. Across every model scale and every benchmark, Breadth-Naive outperforms both the GRPO baseline and Depth-Naive, lifting average Pass@1 (Avg@128) by 1.9–3.7 points on AIME24, MATH500, and Olympiad tasks. This advantage is not merely additive: when breadth is combined with depth through DARS-Breadth, the margin widens further. DARS-Breadth reliably beats both Breadth-Naive and the original DARS variants, confirming our central hypothesis—depth and breadth are complementary, not competing, resources. Their synergy is what unlocks the next tier of LLM reasoning gains.

The practical impact is twofold. First, DARS-Breadth secures the highest Pass@1, the metric that matters most for single-shot deployment. Second, it matches the best Pass@128 scores, demonstrating that the breadth-depth collaboration does not sacrifice the upper-bound capability revealed by heavy sampling. Finally, the choice of schedule matters: the HW schedule consistently yields superior Pass@K curves for both breadth and non-breadth training, while maintaining Pass@1 parity with the ET schedule, making it the preferred option across the board.

### 4.3 TRAINING DYNAMICS AND ABLATION STUDY

In this subsection, we further show more training dynamics to illustrate properties of existing RLVR methods and the superiority of our DARS and DARS-B.

Table 1: Overall performance of *Pass@1* (*Avg@128*) and *Pass@128* of Qwen2.5-Math series.

| Base Model + Method | AIME24 | MATH500 | Olympiad | AMC | Minerva | *Avg@128* | *Pass@128* |
|---|---|---|---|---|---|---|---|
| Qwen2.5-Math-1.5B | 4.0 | 35.1 | 16.2 | 20.8 | 9.5 | 21.1 | 77.9 |
| RLVR baseline | 14.7 | 75.9 | 39.4 | 47.5 | 31.2 | 49.6 | 79.6 |
| Depth-Naive | 16.5 | 76.2 | 39.9 | 47.9 | 30.9 | 50.1 | 79.9 |
| Breadth-Naive | 18.5 | 77.6 | 41.7 | 49.8 | **31.9** | 51.5 | 79.2 |
| **DARS-1.5B-ET** | 15.8 | 76.0 | 40.9 | 47.2 | 30.0 | 50.0 | 81.2 |
| **DARS-1.5B-ET-Breadth** | 18.6 | **79.4** | **42.9** | 50.6 | 31.7 | **52.5** | 80.8 |
| **DARS-1.5B-HW** | 17.7 | 76.4 | 40.0 | 48.4 | 30.8 | 50.0 | 82.1 |
| **DARS-1.5B-HW-Breadth** | **19.3** | 79.0 | 42.7 | **51.9** | 31.6 | 52.4 | **82.2** |
| Qwen2.5-Math-7B | 11.6 | 52.3 | 19.7 | 35.2 | 15.3 | 30.1 | 82.1 |
| RLVR baseline | 26.8 | 82.2 | 44.3 | 57.2 | 35.7 | 55.3 | 81.4 |
| Depth-Naive | 28.0 | 83.8 | 46.4 | 59.0 | 37.3 | 57.0 | 80.3 |
| Breadth-Naive | 30.5 | 83.7 | 47.3 | 61.4 | 37.7 | 57.7 | 79.2 |
| **DARS-7B-ET** | 26.9 | 83.2 | 46.6 | 57.3 | **38.5** | 57.0 | 81.7 |
| **DARS-7B-ET-Breadth** | **33.3** | 83.8 | 47.8 | 61.3 | 38.4 | 58.1 | 82.1 |
| **DARS-7B-HW** | 30.1 | 83.5 | 47.1 | 59.4 | 37.2 | 57.3 | **83.5** |
| **DARS-7B-HW-Breadth** | 33.0 | **84.5** | **48.4** | **63.0** | 36.9 | **58.4** | 83.4 |

***Pass@128* performance surpasses the base model, peaks quickly, and then declines.** We conduct RLVR experiments with rollout size 8/32 to compare our DARS (with $N^{\max} = 32$), the training dynamics of *Pass@128* performance during training is shown in Figure 6. Across all settings, *Pass@128* surpasses the base model during training, but declines after peaking, indicating that overtraining with RLVR harms *Pass@128* performance. Notably, DARS (with $N^{\max} = 32$) incurs substantially less inference cost than naively scaling the rollout size to n = 32. Despite this being an unfair comparison in terms of computational expenditure, our DARS not only attains the highest peak *Pass@128* performance but also outperforms all other settings.

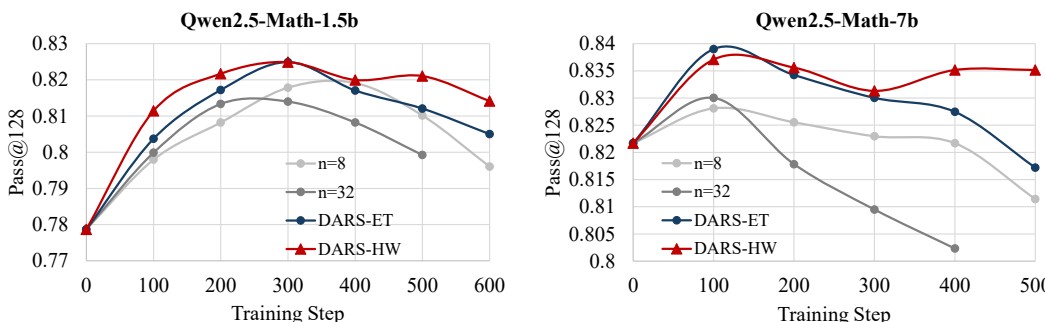

Figure 6: Training dynamics of *Pass@128* performance with different training steps of Qwen2.5-Math-1.5b and Qwen2.5-Math-7b.

**Depth Training with DARS Improve *Pass@K* Performance and Training Efficiency.** Because the *Pass@K* (K=32/128) metric is hard to improve monotonically—it even starts to drop after prolonged training—while *Pass@1* remains comparatively stable and rarely collapses, we seek to boost *Pass@K* without degrading *Pass@1*. Figure 7 plots *Pass@128* against *Pass@1* under a variety of experimental settings. It shows that, at any fixed *Pass@1* level, our DARS method delivers a consistently higher *Pass@128* than the other settings.

Table 2: Average rollout numbers per prompt.

| Model | Naive | DARS-ET | DARS-HW |
|---|---|---|---|
| Qwen2.5-Math-1.5B | 32 | 15.2 (↓**52.5**%) | 23.9 (↓**25.3**%) |
| Qwen2.5-Math-7B | 32 | 12.8 (↓**60.0**%) | 20.1 (↓**37.2**%) |

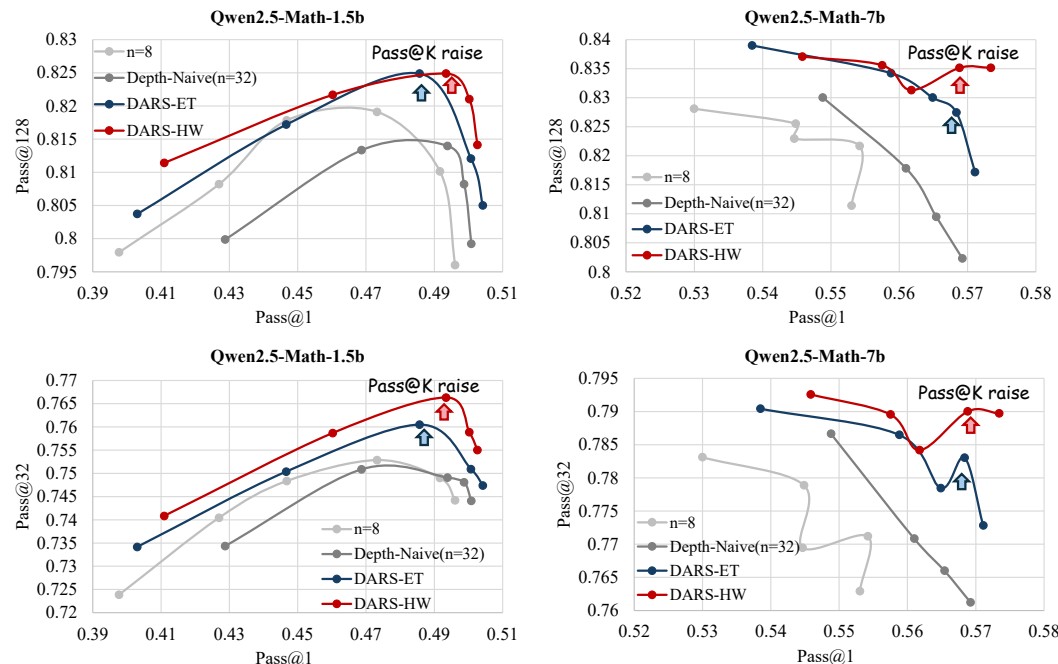

Figure 7: Training dynamics of *Pass@32/Pass@128* and *Pass@1* performance with different training steps of Qwen2.5-Math-1.5b and Qwen2.5-Math-7b.

It is worth noting that, unlike the naive approach of simply increasing the rollout size to 32, our DARS achieves significantly higher training efficiency by allocating more rollouts to the hard problems. As shown in Table 2, our DARS methods need far fewer rollouts than the Depth-Naive method while achieve better performance.

**Depth-Breadth are Complementary in RLVR.** We show that Depth and Breadth are two complementary dimensions in RLVR. As shown in Figure 8, we present the *Pass@1–Pass@K* training-dynamics curves for the Breadth, Depth, and the two-dimensional synergy approach DARS-Breadth. The farther the *Pass@1–Pass@K* curve deviates outward, the more powerful the method. Our DARS-Breadth curves lie on the outermost envelope: it not only achieves the best *Pass@1*, but also simultaneously lifts *Pass@K*. This demonstrates the complementary roles of Depth and Breadth.

**Ablation Study on Base Model.** We further illustrate the effectiveness of DARS on Llama-3.1-8B. The results are shown in Table 3. Our DARS-ET-Breadth achieves both *Pass@1* and *Pass@128* performance compared to other baselines, which further illustrates the effectiveness of our method.

Table 3: Overall performance of *Pass@1* (*Avg@128*) and *Pass@128* performance of Llama-3.1-8B.

| Base Model + Method | AIME24 | MATH500 | Olympiad | AMC | Minerva | *Avg@128* | *Pass@128* |
|---|---|---|---|---|---|---|---|
| Llama-3.1-8B | 0.23 | 6.13 | 1.54 | 2.76 | 2.72 | 3.25 | 52.7 |
| GRPO baseline | 0.66 | 29.6 | 7.09 | 10.1 | 15.7 | 15.8 | 56.5 |
| Depth-Naive | 0.43 | 33.6 | 9.40 | 12.3 | 19.7 | 18.9 | 58.6 |
| Breadth-Naive | 0.79 | 34.4 | 9.34 | 12.2 | 19.0 | 19.0 | 61.1 |
| **DARS-Llama-ET-Breadth** | **1.46** | **39.4** | **12.0** | 13.2 | **20.1** | **22.0** | 67.2 |
| **DARS-Llama-HW-Breadth** | 1.11 | 39.0 | **12.0** | **13.3** | 19.8 | 21.8 | **68.7** |

**Complete Pass@K Accuracy Curve.** We show the complete *Pass@K* curve for Llama-3.1-8B, Qwen2.5-Math-1.5B, and Qwen2.5-Math-7B in Figure 9. The 3 chosen models of DARS are: DARS-Llama-ET-Breadth, DARS-1.5B-HW-Breadth, and DARS-7B-HW-Breadth. DARS models demonstrate a breakthrough in the reasoning boundaries of the base model, especially on the LLama-3.1-8B model, where the improvement in Pass@k is particularly significant.

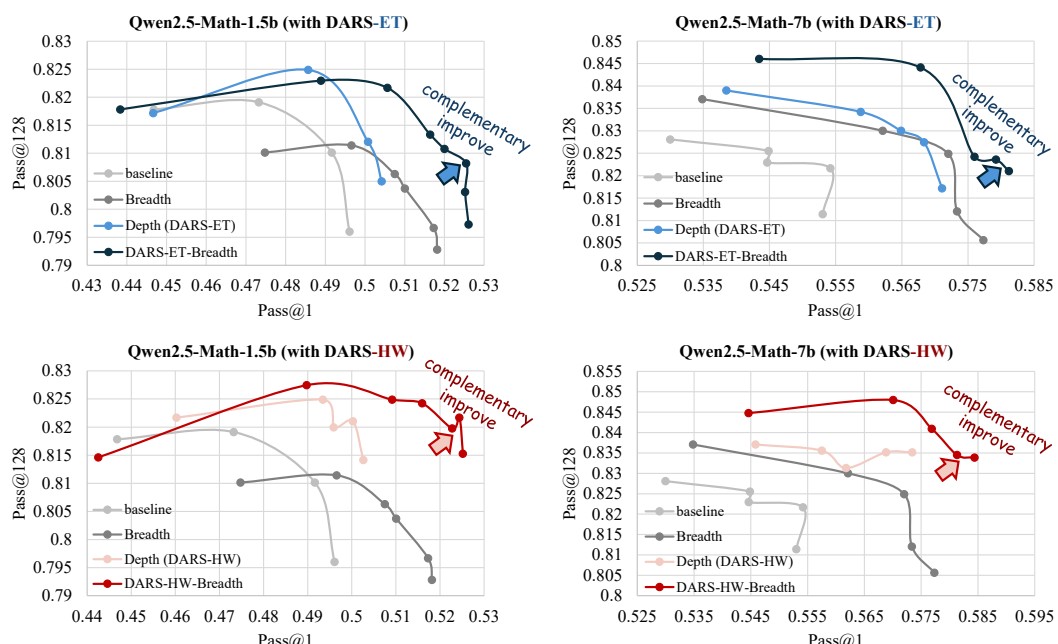

Figure 8: Depth and Breadth Synergy for *Pass@1* and *Pass@K* (K=128) performance.

## 5 RELATED WORKS

Reinforcement Learning (RL) is now standard in post-training LLMs. After early reward-model pipelines (Ouyang et al., 2022b), Direct Preference Optimization (Rafailov et al., 2023) streamlined training by exploiting pairwise preferences. RL with verifiable rewards (RLVR) has since pushed reasoning benchmarks in math and code, culminating in OpenAI's o1 (Jaech et al., 2024) and the zero-RL breakthrough of DeepSeek-R1 (Guo et al., 2025). Follow-up Large Reasoning Models—Kimi 1.5 (Team et al., 2025), Gemini-Think (DeepMind, 2024), QwQ (Qwen, 2024)—and studies like Zeng et al. (2025); Luo et al. (2025) further validate RLVR. The leading algorithm, GRPO (Shao et al., 2024), extends PPO (Schulman et al., 2017) with group-relative advantages, inspiring DAPO (Yu et al., 2025), VAPO (Yue et al., 2025b), and Dr. GRPO (Liu et al., 2025b). Yet GRPO and its variants systematically undervalue hard problems, hurting Pass@K. More related works are shown in Appendix B.

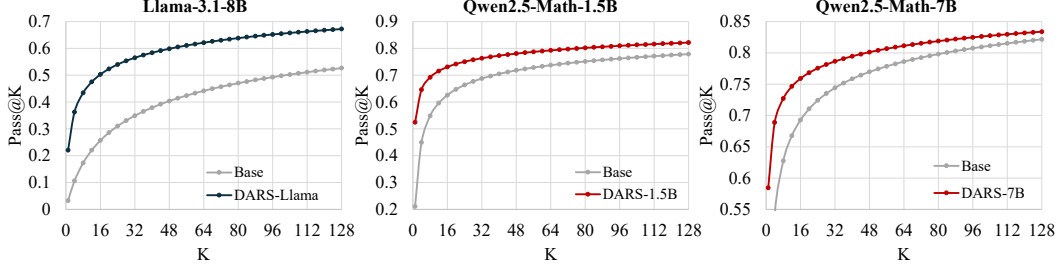

Figure 9: Complete *Pass@K* accuracy curve of base models and our DARS models.

## 6 CONCLUSION

In this work, we reveal that GRPO-based RLVR methods under-weight hard problems due to cumulative-advantage bias, capping *Pass@K*. Our DARS sampler cheaply re-allocates rollouts to these hard instances, while large-breadth training with full-batch updates raises Pass@1. The unified DARS-Breadth framework jointly lifts *Pass@1* and *Pass@K*, proving depth and breadth are synergistic levers in RLVR.

## 7 REPRODUCIBILITY STATEMENT

We have included a comprehensive reproducibility package as part of our supplementary materials to facilitate the replication of all experiments presented in this paper. This includes anonymized source code implementing the proposed model and training procedures, as well as the preprocessed datasets used in our experiments. Detailed instructions for environment setup, data preparation, and execution are provided in the accompanying README documentation. Furthermore, we have supplied exact configuration files and scripts specifying all hyperparameters, and training commands required to reproduce our results.

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

## APPENDIX

## A THE USE OF LARGE LANGUAGE MODELS

During manuscript preparation, a large language model (LLM) was occasionally employed as an auxiliary assistant to refine language expression, such as improving sentence fluency and enhancing readability. The model was not involved in generating original research contributions: it did not participate in formulating research questions, designing methodologies, conducting experiments, analyzing results, or drafting substantive scientific content. All core intellectual work, including the development of ideas, execution of experiments, and interpretation of findings, was carried out independently by the authors. Any linguistic suggestions offered by the LLM were critically reviewed

and selectively incorporated, ensuring that accuracy, originality, and scholarly integrity were fully maintained. The authors alone bear responsibility for the research content and conclusions, and the LLM is not listed as a contributor or author.

## B  MORE RELATED WORKS

With the rapid advancement of RLVR and the proliferation of open-source LRMs, many studies have begun to analyze the RLVR pipeline and these open LRMs. Several studies (Liu et al., 2025a; Zhao et al., 2025; Shah et al., 2025) indicates that the self-reflect and self-critique behaviors observed after RLVR originates from the base model rather than the RL process. Dang et al. (2025) find that although the RLVR process benefits *Pass@1*, *Pass@K* may decline as training progresses. Subsequently, Yue et al. (2025a) through extensive experimental analysis, discovered that RLVR's performance is significantly constrained by the base model; once training converges, it struggles to surpass the capability boundary of the base model. These studies have sparked widespread concern about the capability ceiling of RLVR, and consequently, the *Pass@K* metric has become a focal point for diagnosing and potentially transcending the intrinsic limits imposed by the base model (Liang et al., 2025). This paper analyzes and refines the RLVR pipeline from the dual perspectives of *Pass@1* and *Pass@K*.

## C  DERIVATION OF ADDITIONAL ROLLOUTS $\Delta n_j$

The cumulative advantage for a group with accuracy $\hat{a}_j$ and total rollout size $N_j = N^{pre} + \Delta n_j$ is given by:

$$\mathcal{A}_{\text{group}}(\hat{a}_j, N_j) = N_j \cdot \mathcal{S}(\hat{a}_j),$$

where $\mathcal{S}(\hat{a}_j) = 2\hat{a}_j(1 - \hat{a}_j)$.

After the first-stage rollout of size $N^{pre}$, the initial cumulative advantage is:

$$\mathcal{A}_{\text{group}}^{N^{pre}}(\hat{a}_j) = N^{pre} \cdot \mathcal{S}(\hat{a}_j).$$

Our goal is to determine the number of additional trajectories $\Delta n_j$ needed so that the final cumulative advantage $\mathcal{A}_{\text{group}}(\hat{a}_j, N_j)$ meets a target value $\mathcal{A}_{\text{group}}^{\text{target}}(\hat{a}_j)$.

**Equal-Treatment (ET) Schedule:**

The target cumulative advantage is set to be constant for all questions with $\hat{a}_j < 0.5$:

$$\mathcal{A}_{\text{group}}^{\text{ET}}(\hat{a}_j) = \mathcal{A}_{\text{group}}^{N^{pre}}(0.5) = N^{pre} \cdot \mathcal{S}(0.5).$$

We solve for $\Delta n_j^{\text{ET}}$:

$$\mathcal{A}_{\text{group}}(\hat{a}_j, N_j) = \mathcal{A}_{\text{group}}^{\text{ET}}(\hat{a}_j)$$
$$(N^{pre} + \Delta n_j^{\text{ET}}) \cdot \mathcal{S}(\hat{a}_j) = N^{pre} \cdot \mathcal{S}(0.5)$$
$$\Delta n_j^{\text{ET}} \cdot \mathcal{S}(\hat{a}_j) = N^{pre} \cdot \mathcal{S}(0.5) - N^{pre} \cdot \mathcal{S}(\hat{a}_j)$$
$$\Delta n_j^{\text{ET}} = \frac{N^{pre} \cdot \mathcal{S}(0.5) - N^{pre} \cdot \mathcal{S}(\hat{a}_j)}{\mathcal{S}(\hat{a}_j)}.$$
$$\Delta n_j^{\text{ET}} = \frac{\mathcal{A}_{\text{group}}^{N^{pre}}(0.5) - \mathcal{A}_{\text{group}}^{N^{pre}}(\hat{a}_j)}{\mathcal{S}(\hat{a}_j)}.$$

The rollout size must be an integer, and we cap the total rollout sampling upper limit at $N^{max}$, so

$$\Delta n_j^{\text{ET}} = \min\left(\left\lceil \frac{\mathcal{A}_{\text{group}}^{N^{pre}}(0.5) - \mathcal{A}_{\text{group}}^{N^{pre}}(\hat{a}_j)}{\mathcal{S}(\hat{a}_j)} \right\rceil, N^{\max} - N^{pre}\right).$$

**Hardness-Weighted (HW) Schedule:**

The target cumulative advantage increases with difficulty:

$$\mathcal{A}_{\text{group}}^{\text{HW}}(\hat{a}_j) = 2(1 - \hat{a}_j) \cdot \mathcal{A}_{\text{group}}^N(0.5) = 2x_j \cdot N^{pre} \cdot \mathcal{S}(0.5).$$

We solve for $\Delta n_j^{\text{HW}}$:

$$\mathcal{A}_{\text{group}}(\hat{a}_j, N_j) = \mathcal{A}_{\text{group}}^{\text{HW}}(\hat{a}_j)$$

$$(N^{pre} + \Delta n_j^{\text{HW}}) \cdot \mathcal{S}(\hat{a}_j) = 2x_j \cdot N^{pre} \cdot \mathcal{S}(0.5)$$

$$\Delta n_j^{\text{HW}} \cdot \mathcal{S}(\hat{a}_j) = 2x_j \cdot N^{pre} \cdot \mathcal{S}(0.5) - N^{pre} \cdot \mathcal{S}(\hat{a}_j)$$

$$\Delta n_j^{\text{HW}} = \frac{2x_j \cdot N^{pre} \cdot \mathcal{S}(0.5) - N^{pre} \cdot \mathcal{S}(\hat{a}_j)}{\mathcal{S}(\hat{a}_j)}.$$

Again, using the baseline advantage notation $\mathcal{A}_{\text{group}}^{N^{pre}}(\hat{a}_j) = N^{pre} \cdot \mathcal{S}(\hat{a}_j)$, we obtain:

$$\Delta n_j^{\text{HW}} = \min\left(\left\lceil \frac{2x_j \cdot \mathcal{A}_{\text{group}}^{N^{pre}}(0.5) - \mathcal{A}_{\text{group}}^{N^{pre}}(\hat{a}_j)}{\mathcal{S}(\hat{a}_j)} \right\rceil, N^{\max} - N^{pre}\right).$$

Both derivations include a ceiling function and are capped at $N^{\max}$ to control computational cost, as shown in Equations 6 and 8 in the paper.

# D   MATHEMATICAL DERIVATION: GROUP CUMULATIVE ADVANTAGE AND GRPO GRADIENT NORM

In this appendix, we provide a detailed mathematical derivation demonstrating the relationship between Group Cumulative Advantage and the gradient norm in GRPO. This derivation substantiates the claim that $\mathcal{A}_{\text{group}}$ serves as an effective indicator of the model's attention to specific problems during training.

## D.1   GRPO GRADIENT FORMULATION

The GRPO objective function and its gradient are given by (we can remove the clip operation for simplified analysis):

$$\mathcal{J}_{\text{GRPO}}(\theta) = \mathbb{E}_{q \sim \mathcal{D}, \{o_i\}_{i=1}^G \sim \pi_{\theta_{old}}(\cdot|q)}\left[\frac{1}{\sum_{i=1}^G |o_i|} \sum_{i=1}^G \sum_{t=1}^{|o_i|} \rho_{i,t}(\theta)\hat{A}_i\right], \tag{11}$$

$$\nabla_\theta \mathcal{J}_{\text{GRPO}}(\theta) = \mathbb{E}_{q \sim \mathcal{D}, \{o_i\}_{i=1}^G \sim \pi_{\theta_{old}}(\cdot|q)}\left[\frac{1}{\sum_{i=1}^G |o_i|} \sum_{i=1}^G \sum_{t=1}^{|o_i|} \hat{A}_i \nabla_\theta \rho_{i,t}(\theta)\right]$$

$$= \mathbb{E}_{q \sim \mathcal{D}, \{o_i\}_{i=1}^G \sim \pi_{\theta_{old}}(\cdot|q)}\left[\frac{1}{\sum_{i=1}^G |o_i|} \sum_{i=1}^G \sum_{t=1}^{|o_i|} \hat{A}_i \frac{\nabla_\theta \pi_\theta(o_{i,t}|q, o_{i,<t})}{\pi_{\theta_{old}}(o_{i,t}|q, o_{i,<t})}\right]$$

$$= \mathbb{E}_{q \sim \mathcal{D}, \{o_i\}_{i=1}^G \sim \pi_{\theta_{old}}(\cdot|q)}\left[\frac{1}{\sum_{i=1}^G |o_i|} \sum_{i=1}^G \sum_{t=1}^{|o_i|} \rho_{i,t}(\theta)\hat{A}_i \nabla_\theta \log \pi_\theta(o_{i,t}|q, o_{i,<t})\right]. \tag{12}$$

## D.2   DERIVATION OF GRADIENT NORM UPPER BOUND

We now derive the upper bound relationship between the gradient norm and Group Cumulative Advantage.

The gradient norm of our adopted GRPO algorithm is shown as the following:

$$\|\nabla_\theta \mathcal{J}_{\text{GRPO}}(\theta)\| = \left\|\mathbb{E}\left[\frac{1}{\sum_{i=1}^G |o_i|} \sum_{i=1}^G \sum_{t=1}^{|o_i|} \rho_{i,t}(\theta)\hat{A}_i \nabla_\theta \log \pi_\theta(o_{i,t}|q, o_{i,<t})\right]\right\|. \tag{13}$$

Since the norm is a convex function and expectation is linear, by Jensen's inequality:
$$\|\mathbb{E}[X]\| \leq \mathbb{E}[\|X\|] \tag{14}$$

Thus:
$$\|\nabla_\theta \mathcal{J}_{\text{GRPO}}(\theta)\| \leq \mathbb{E}\left[\left\|\frac{1}{\sum_{i=1}^G |o_i|} \sum_{i=1}^G \sum_{t=1}^{|o_i|} \rho_{i,t}(\theta)\hat{A}_i \nabla_\theta \log \pi_\theta(o_{i,t}|q, o_{i,<t})\right\|\right]. \tag{15}$$

Applying the triangle inequality to the inner summation:
$$\left\|\sum_{i=1}^G \sum_{t=1}^{|o_i|} a_{i,t}\right\| \leq \sum_{i=1}^G \sum_{t=1}^{|o_i|} \|a_{i,t}\|, \tag{16}$$

where:
$$a_{i,t} = \frac{1}{\sum_{i=1}^G |o_i|} \rho_{i,t}(\theta)\hat{A}_i \nabla_\theta \log \pi_\theta(o_{i,t}|q, o_{i,<t}). \tag{17}$$

Therefore:
$$\|\nabla_\theta \mathcal{J}_{\text{GRPO}}(\theta)\| \leq \mathbb{E}\left[\frac{1}{\sum_{i=1}^G |o_i|} \sum_{i=1}^G \sum_{t=1}^{|o_i|} \left\|\rho_{i,t}(\theta)\hat{A}_i \nabla_\theta \log \pi_\theta(o_{i,t}|q, o_{i,<t})\right\|\right]. \tag{18}$$

Then we have,
$$\left\|\rho_{i,t}(\theta)\hat{A}_i \nabla_\theta \log \pi_\theta(o_{i,t}|q, o_{i,<t})\right\| = |\rho_{i,t}(\theta)| \cdot |\hat{A}_i| \cdot \|\nabla_\theta \log \pi_\theta(o_{i,t}|q, o_{i,<t})\|. \tag{19}$$

Thus:
$$\|\nabla_\theta \mathcal{J}_{\text{GRPO}}(\theta)\| \leq \mathbb{E}\left[\frac{1}{\sum_{i=1}^G |o_i|} \sum_{i=1}^G \sum_{t=1}^{|o_i|} |\rho_{i,t}(\theta)| \cdot |\hat{A}_i| \cdot \|\nabla_\theta \log \pi_\theta(o_{i,t}|q, o_{i,<t})\|\right]. \tag{20}$$

We further take a boundedness assumption. in policy optimization, we assume the gradient log-probabilities $\|\nabla_\theta \log \pi_\theta(o_{i,t}|q, o_{i,<t})\|$ are bounded. Furthermore, it's vital to notice that the importance ratio $\rho_{i,t}(\theta)$ are also bounded through the clip operation in GRPO algorithm.

Thus, there exists a constant $C > 0$ such that:
$$|\rho_{i,t}(\theta)| \cdot \|\nabla_\theta \log \pi_\theta(o_{i,t}|q, o_{i,<t})\| \leq C. \tag{21}$$

Therefore:
$$\|\nabla_\theta \mathcal{J}_{\text{GRPO}}(\theta)\| \leq C \cdot \mathbb{E}\left[\frac{1}{\sum_{i=1}^G |o_i|} \sum_{i=1}^G \sum_{t=1}^{|o_i|} |\hat{A}_i|\right]. \tag{22}$$

Noting that $|\hat{A}_i|$ is independent of $t$ for fixed $i$:
$$\sum_{i=1}^G \sum_{t=1}^{|o_i|} |\hat{A}_i| = \sum_{i=1}^G |o_i| \cdot |\hat{A}_i|. \tag{23}$$

Thus:
$$\|\nabla_\theta \mathcal{J}_{\text{GRPO}}(\theta)\| \leq C \cdot \mathbb{E}\left[\frac{\sum_{i=1}^G |o_i| \cdot |\hat{A}_i|}{\sum_{i=1}^G |o_i|}\right]. \tag{24}$$

In this paper, we define the Group Cumulative Advantage as:
$$\mathcal{A}_{\text{group}} = \sum_{i=1}^G |\hat{A}_i|. \tag{25}$$

By the weighted arithmetic mean inequality:
$$\frac{\sum_{i=1}^G |o_i| \cdot |\hat{A}_i|}{\sum_{i=1}^G |o_i|} \leq \frac{\sum_{i=1}^G |o_i| \cdot \sum_{i=1}^G |\hat{A}_i|}{\sum_{i=1}^G |o_i|} = \sum_{i=1}^G |\hat{A}_i| = \mathcal{A}_{\text{group}}. \tag{26}$$

Therefore:
$$\|\nabla_\theta \mathcal{J}_{\text{GRPO}}(\theta)\| \leq C \cdot \mathbb{E}[\mathcal{A}_{\text{group}}]. \tag{27}$$

### D.3 Theoretical Implications

This derivation establishes that Group Cumulative Advantage $\mathcal{A}_{\text{group}}$ provides an upper bound for the expected gradient norm in GRPO. Consequently: Larger $\mathcal{A}_{\text{group}}$ values indicate stronger gradient signals for the corresponding problem. The training process allocates more "attention" to problems with higher $\mathcal{A}_{\text{group}}$ values during parameter updates. Therefore, we consider $\mathcal{A}_{\text{group}}$ serves as a mathematically grounded indicator of problem importance in GRPO training dynamics.

This theoretical foundation validates the use of Group Cumulative Advantage as a meaningful metric for analyzing training behavior and problem prioritization in GRPO.

## E Training and Evaluation Details

> **Prompt for Solving Complex Reasoning Tasks**
>
> Your task is to solve the given question step by step. You should conduct a systematic, thorough reasoning process before providing the final answer. This involves analyzing, summarizing, exploring, reassessing, and refining your reasoning process through multiple iterations. Each reasoning step should include detailed analysis, brainstorming, verification, and refinement of ideas. You should include the final answer in \boxed{} for closed-form results like multiple choices or mathematical results.

**Parameters and Metrics.** Currently, our experiments are conducted with Qwen2.5-Math series language models (Yang et al., 2024). We set the temperature to 1.0 for both the training and evaluation procedures. In this paper, we mainly use two metrics, *Pass@1* and *Pass@K*. To acquire *Pass@K* results, we sample 128 candidate responses for each question during the evaluation process; the calculation of *Pass@1* is derived from *Avg@128*. Both the training and evaluation processes are scored using Math-Verify. The learning rate is 1e-6 for depth training methods, and 5e-6 for large breadth training. We do not use the reference model and KL loss. For fair comparison, we uniformly set the PPO mini step to 2 for all experiments. By default, the maximum prompt length is 1024, and the maximum response length is 3072 for the Qwen2.5-Math series model.

Moreover, we have adopted the same unbiased, low-variance estimator for *pass@k* as used in prior works (Yue et al., 2025a; Chen et al., 2021),

$$\text{pass@}K = \mathbb{E}_{x_i \sim \mathcal{D}} \left[ 1 - \frac{\binom{n-c_i}{k}}{\binom{n}{k}} \right],$$

Specifically, when $K = N$, the metric become: $\text{pass@}K = c_1 \vee c_2 \vee \cdots c_{128}$.

**Implementation Details.** Following LUFFY (Yan et al., 2025), we use the default subset and filter out generations that are longer than 8192 tokens and those that are verified wrong by Math-Verify [1], resulting in 45k question-solution pairs. For training Llama-3.1-8B, we use the train split of MATH dataset. Our training framework is derived from Verl (Sheng et al., 2024) pipeline, which is a flexible, high-performance reinforcement-learning framework built for training large language-model agents. With native PyTorch support and efficient distributed training, Verl lets researchers quickly prototype and scale RL algorithms like PPO on GPUs. Following Dr. GRPO (Liu et al., 2025b), we remove the KL loss and the length normalization in GRPO. All of our experiments are conducted on H200 GPUs. At present, the LLM of our experiment is the Qwen2.5-Math series.

**Training Steps and Checkpoint Steps.** For non-breadth methods on Qwen2.5-Math-1.5B/7B, we set the checkpoint step as 100. For breadth methods on Qwen2.5-Math-1.5B/7B, we set the checkpoint step as 15. The specific training steps are determined according to the convergence of the model. The number of training steps for non-breadth training is set as 300 for Llama-3.1-8B, 600 for Qwen2.5-Math-1.5B, and 500 for Qwen2.5-Math-7B. The number of training steps for breadth training is set as 70. For breadth training, we set the total training steps as 105 for Qwen2.5-Math-1.5B, and 75 for Qwen2.5-Math-7B.

---

[1] https://github.com/huggingface/Math-Verify

# F MORE EXPERIMENTAL RESULTS

## F.1 ABLATION STUDY ON STD-BASED ADVANTAGE COMPUTATION

As illustrated in Section 2, Dr. GRPO (Liu et al., 2025b) removes the standard-deviation term from the advantage computation to eliminate question-level difficulty bias, and demonstrates its superiority through extensive experiments. Consequently, the experiments reported in this study were conducted primarily though the Dr. GRPO methodology. To further illustrate the effectiveness of DARS on std-based advantage computation, we conduct the experiment with HW schedule on Qwen2.5-Math-1.5B model, as shown in Figure 10

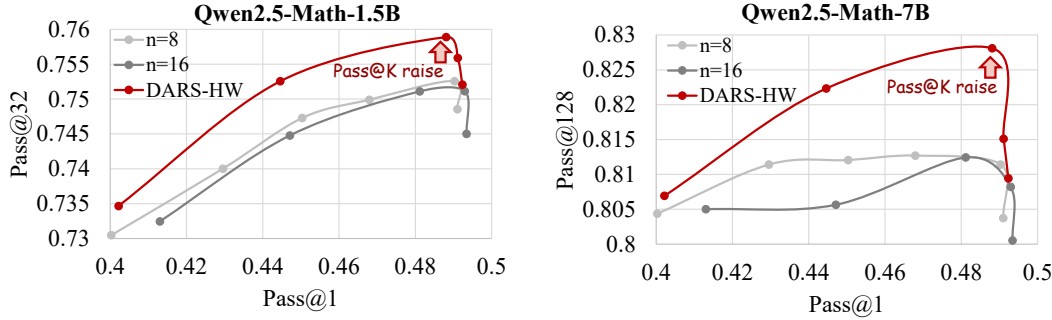

Figure 10: Comparison of our DARS on std-based advantage computation.

## F.2 DEPTH AND BREADTH SYNERGY FOR PASS@1 AND PASS@32

In Section 4.3, we show the training dynamics of *Pass@128-Pass@1* for DARS and baseline methods. To further illustrate the effectiveness of DARS, we show the training dynamics of *Pass@32-Pass@1* in Figure 11. Our DARS significantly improves the *Pass@32* performance compared to other methods.

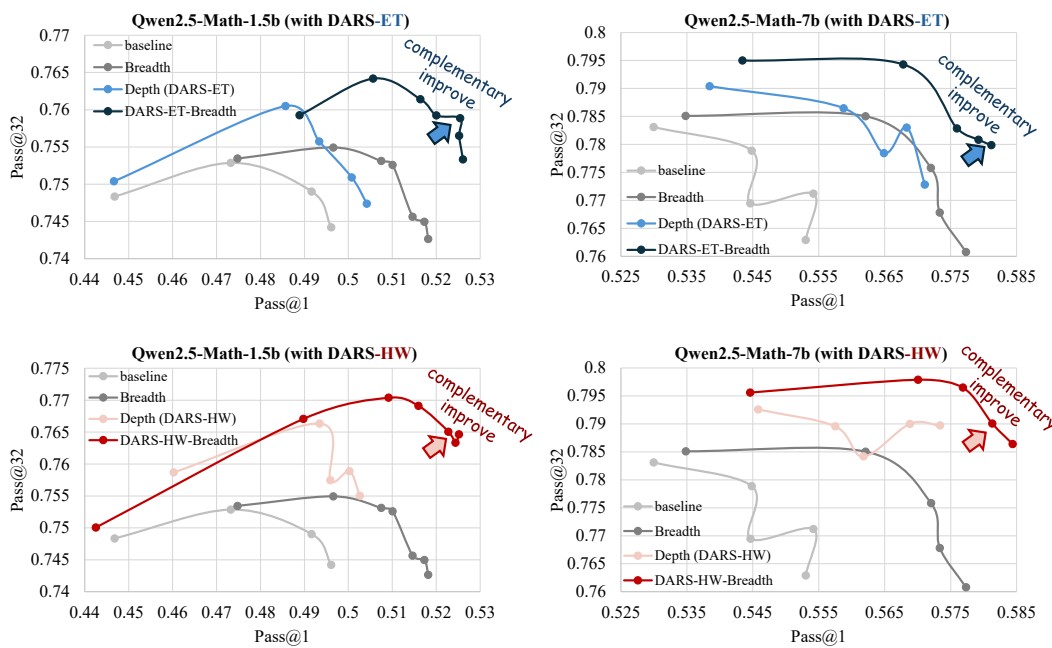

Figure 11: Complementary improve of Depth and Breadth Synergy for *Pass@1* and *Pass@K* (K=32) performance.

### F.3 COMPARISON OF ET/HW SCHEDULE IN BREADTH SCALING

In addition, compared with the ET schedule, DARS-HW-Breadth significantly improves the model's Pass@128 performance as shown in Figure 12. We consider this performance gain is due to the HW schedule placing greater emphasis on difficult samples.

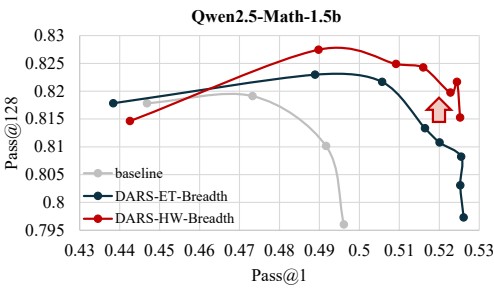 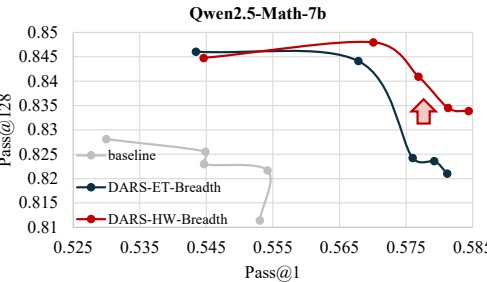

Figure 12: Comparison of ET and HW schedule in breadth training of Qwen2.5-Math series.

### F.4 IMPACT OF TEMPERATURE

Some researches (Karan & Du, 2025; Qin et al., 2025; Ni et al., 2025) indicates that temperature matters in LLM reasoning. To further illustrate the performance improvement under different temperature, we additionally add the above experiments. The results shows that the improvement of our method is consistent over different temperature. The results are shown in Figure 13.

| | Qwen2.5-Math-1.5B | | | | | | Qwen2.5-Math-7B | | | | |
| --- | --- | --- | --- | --- | --- | --- | --- | --- | --- | --- | --- |
| | t=0.6 | t=0.8 | t=1.0 | t=1.2 | t=1.4 | | t=0.6 | t=0.8 | t=1.0 | t=1.2 | t=1.4 |
| GRPO-Baseline | 78.9 | 79.2 | 79.6 | 80 | 80 | GRPO-Baseline | 80.4 | 80.7 | 81.4 | 81.6 | 81.6 |
| Depth-Naive | 79.4 | 79.6 | 79.9 | 80.3 | 80.4 | Depth-Naive | 79.7 | 80 | 80.3 | 80.7 | 80.8 |
| Breadth-Naive | 78.9 | 78.9 | 79.2 | 79.6 | 79.9 | Breadth-Naive | 77.7 | 78.1 | 79.2 | 79.4 | 79.6 |
| DARS-HW-Breadth | **81.6** | **81.9** | **82.2** | **82.6** | **82.7** | DARS-HW-Breadth | **82.6** | **82.8** | **83.4** | **84.0** | **84.0** |

Figure 13: Heat map of *Pass@128* of Qwen2.5-Math series in different temperatures.

### F.5 CONSISTENT IMPROVEMENT DURING RL PROCESS

To further show that our method consistently improve model performance, we calculated the mean of Pass@128 and Pass@32 for the last 3 checkpoints of each method, as shown in Table 4.

### F.6 PERFORMANCE OF NONE MATH MODEL.

We further evaluate our method on Qwen2.5-Math-7B-Instruct. The results are shown in Table 5. As the results show, our method still outperforms the baseline in both the Pass@1 and Pass@K metrics.

### F.7 DARS ELICITS LONGER REASONING CHAINS

This section investigates how DARS influences the reasoning length of LLMs. We tracked the response length dynamics during the training of Qwen2.5-Math-1.5B and 7B models. Our experiments reveal two key observations: (1) The training process shows a clear trend of increasing generation length, as shown in Figure 14. (2) When evaluated on AIME 2024, models trained with DARS consistently produce longer reasoning traces than the baseline, as shown in Figure 15. These results

Table 4: Average performance of *Pass@1/32/128* for the last 3 checkpoints during training.

| Model | Pass@1 | Pass@32 | Pass@128 |
|---|---|---|---|
| *Qwen2.5-Math-1.5B as the Base Model* | | | |
| GRPO-Baseline | 48.8 | 74.7 | 80.8 |
| Depth-Naive | 49.5 | 74.4 | 80.6 |
| Breadth-Naive | 51.4 | 74.4 | 79.8 |
| DARS-HW | 49.5 | 75.7 | 81.9 |
| **DARS-HW-Breadth** | **52.4** | **76.4** | **82.1** |
| *Qwen2.5-Math-7B as the Base Model* | | | |
| GRPO-Baseline | 55.1 | 76.9 | 81.8 |
| Depth-Naive | 56.4 | 76.7 | 80.9 |
| Breadth-Naive | 57.2 | 76.7 | 81.3 |
| DARS-HW | 56.8 | 78.8 | 83.4 |
| **DARS-HW-Breadth** | **58.3** | **79.1** | **83.7** |

Table 5: Overall performance of *Pass@1* (*Avg@128*) and *Pass@128* of Qwen2.5-7B-Instruct.

| Model | AIME24 | Math500 | Olympiad | AMC | Minerva | Avg@128 | Pass@128 |
|---|---|---|---|---|---|---|---|
| Qwen2.5-7B-Instruct | 11.9 | 72.3 | 37.1 | 42.2 | 31.9 | 47.2 | 80.3 |
| GRPO-baseline | 14.2 | 74.8 | 37.6 | 43.4 | 33.4 | 48.6 | 78.8 |
| **DARS-HW-Breadth** | 15.6 | 76.5 | 38.4 | 44.7 | 34.6 | 49.6 | **82.3** |

provide concrete evidence that our DARS method successfully stimulates the model to perform deeper and more thorough thinking.

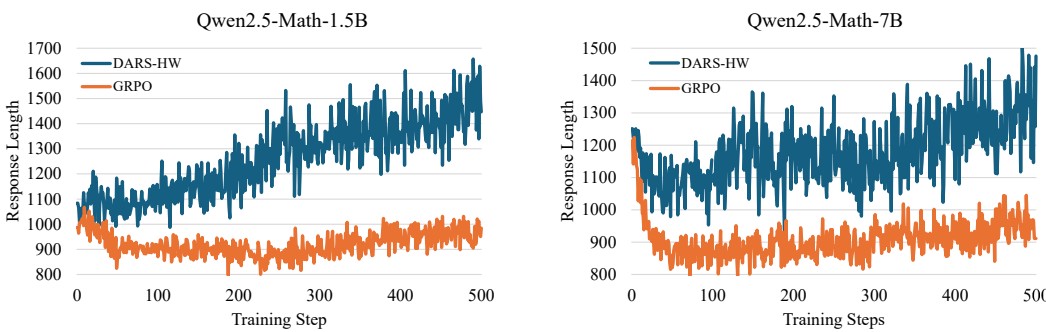

Figure 14: Training dynamics of response length for GRPO and DARS.

## G   DISCUSSION AND FUTURE WORK

In this section, we analyze how hyperparameters $N$ and $N^{\mathrm{max}}$ control the shape of the cumulative advantage curve, and how this shape may influence training behavior. We further discuss how dynamically adjusting these parameters could enable a smooth transition from *Pass@K*-oriented to *Pass@1*-oriented training.

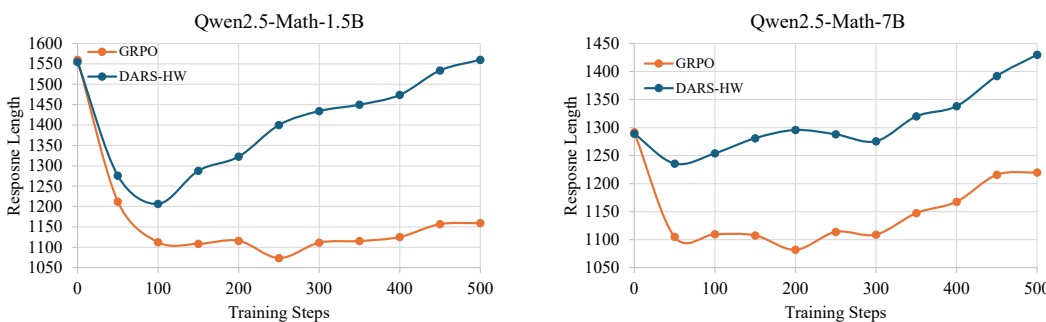

Figure 15: Statistical results of response length on AIME 2024 for GRPO and DARS.

### G.1 HYPERPARAMETER CONTROL OF CUMULATIVE ADVANTAGE SHAPE

We show the Cumulative Advantage shape of ET/HW schedule with $N = 8$ in Figure 16. By continuously reducing the size of $N_{\max}$, the curve will contract accordingly. When $N_{\max} = N$, it is equivalent to the vanilla method without performing DARS.

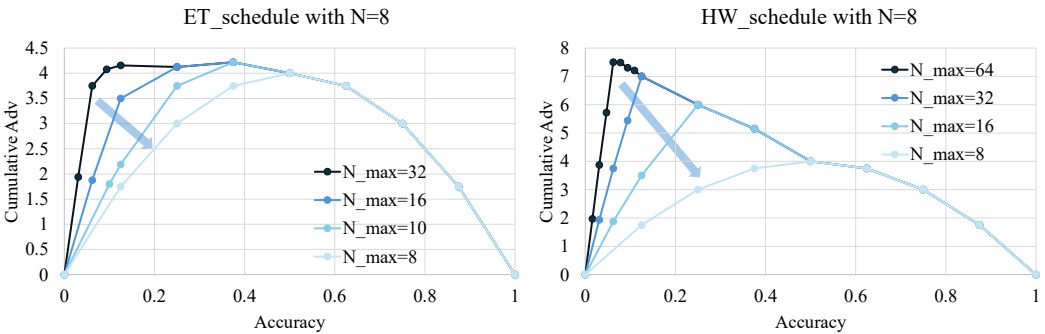

Figure 16: Control the shape of Cumulative Advantage by adjusting the $N_{max}$ hyperparameter of DARS.

### G.2 POTENTIAL PASS@K TO PASS@1 TRAINING TRANSITION

The dynamic control of $N^{\max}$ suggests an intriguing training strategy: starting with a large $N^{\max}$ value to maximize *Pass@K* performance through extensive exploration of hard problems, then gradually reducing $N^{\max}$ throughout training to transition toward *Pass@1* optimization.

This approach mirrors curriculum learning principles, where the training difficulty is progressively adjusted. Initially, the model benefits from the expanded solution space and diverse reasoning patterns discovered through heavy sampling on hard problems (high $N^{\max}$). As training progresses and the model's capability matures, reducing $N^{\max}$ focuses the training on refining the most promising solution strategies, ultimately improving single-shot performance.

Future work will explore optimal annealing schedules for $N^{\max}$ and investigate whether this transition strategy can simultaneously maximize both *Pass@1* and *Pass@K* performance, potentially overcoming the current limitations of RLVR training.

