# OpenReview forum: "Depth-Breadth Synergy in RLVR: Unlocking LLM Reasoning Gains with Adaptive Exploration"
_ICLR.cc/2026/Conference — ICLR 2026 Conference Withdrawn Submission_

### Official Review · Reviewer_wQQe · 2025-10-27

**Soundness:** 3
**Presentation:** 2
**Contribution:** 2
**Rating:** 4
**Confidence:** 3

**Summary:**

This paper analyzes RLVR and argues that existing methods such as GRPO underweight hard problems (depth bias). It proposes Difficulty-Adaptive Rollout Sampling (DARS) to allocate more rollouts to low-accuracy samples and examnines breadth scaling (large-batch updates) as a complementary factor. The authors calim that combining the two (DARS-breadth) yields simultaneous improvements in Pass@1 and Pass@K across math reasoning benchmarks.

**Strengths:**

1. The paper provides the intuition that GRPO (variants) emphasizes medium-difficulty problems and train less on challenging problems, through math derivation by showing A_{group} = 2 N u (1 - u).
2. The paper evaluates the DARS and variants on AIME, OlympiadBench, Minerva, MATH500 with multiple model sizes, showing consistent performance trends.

**Weaknesses:**

1. The performance comparison between GRPO and DARS is not controlled for solution length. Since solution length is generally positively correlated with correctness, and DARS places more emphasis on hard problems that naturally require longer reasoning, it is possible that DARS merely encourages longer generations rather than genuinely improving reasoning ability.
2. The “breadth scaling” results are unsurprising—larger batch sizes often stabilize RL training—and the idea itself is not novel.
3. The claimed synergy between depth and breadth is not convincingly demonstrated. The improvements appear largely additive rather than reflecting a meaningful interaction between the two dimensions.
4. The methodology section is difficult to parse, with inconsistent notation and somewhat heavy presentation. The paper would benefit from clearer mathematical exposition and cleaner notation.

**Questions:**

See weakness

---

> ### Author Response · Authors · 2025-11-21
> **To reviwer wQQe**
>
> ### **1. About Resposne Length**
> Thank you for this insightful comment. We agree that controlling for solution length is crucial for a fair comparison.
>
> We would like to clarify that in our experiments, the **maximum response length was strictly controlled and kept identical** for all methods, including the GRPO baseline and DARS variants (as detailed in our training setup). Therefore, the performance gains from DARS cannot be attributed to a simple allowance for longer generations.
>
> Furthermore, we view the observed extension in reasoning chains not as a confounder but as a positive signal of deeper reasoning, a phenomenon also noted in models like DeepSeek-R1 [1]. Hard problems inherently require more detailed derivations. Our new analysis in Appendix F.7 shows DARS adaptively extends reasoning length, which we interpret as the model engaging in more thorough problem-solving.
>
> The concurrent improvements in `Pass@K` and `Pass@1` suggest these are genuine reasoning gains, not merely a byproduct of verbosity.
>
> [1] *DeepSeek-R1: Incentivizing Reasoning Capability in LLMs via Reinforcement Learning.*
>
> ---
>
> ### **2. About “Breadth Scaling”**
>
> We appreciate the reviewer's point. Our contribution in “Breadth Scaling” lies in the follwoing 2 points:
>
> * **Systematic Analysis**: While larger batch sizes are known to stabilize RL, our contribution lies not in this fact itself, but in systematically revealing its role as implicit entropy regularization that sustains exploration and specifically boosts Pass@1.
> * **Full Batch Update with Unfixed Batch Size**: Morver, our breadth scaling is not merely increasing batch size naively. It replaces PPO’s mini-batch updates with full-batch gradient descent across multiple PPO epochs, which eliminates gradient noise and sustains token-level exploration. Our breadth scaling has adapted to the non-fixed batch size of DARS and is naturally suitable for the combined training of both breadth and depth dimensions.
>
> ---
>
> ### **3. About Synergy Improvement.**
> We respectfully disagree with the reviewer’s assessment. Our results demonstrate that depth (DARS) and breadth (large-batch training) are not merely additive but exhibit clear synergy. While DARS alone boosts Pass@K and breadth enhances Pass@1, their combination in DARS-Breadth yields simultaneous gains in both metrics—surpassing the sum of individual improvements. Figure 8 shows the outermost training envelope, where DARS-Breadth consistently outperforms all depth-only or breadth-only baselines, indicating a complementary interaction: depth expands reasoning boundaries, while breadth stabilizes training and sustains exploration. This is not achievable by simple addition, confirming a meaningful, synergistic interplay between the two dimensions.
>
> ---
>
> ### **4. Clearer Mathematical Exposition and Cleaner Notation.**
>
> Thank you for the feedback. We have revised the methodology section to unify the notation and streamline the mathematical presentation, significantly improving its clarity. The updated manuscript is now much easier to follow.

---

### Official Review · Reviewer_qq1K · 2025-10-30

**Soundness:** 2
**Presentation:** 4
**Contribution:** 2
**Rating:** 2
**Confidence:** 4

**Summary:**

The paper firstly tries to present a bias in GRPO which allocates more attention to medium-difficulty problems. Then, they present DARS: Difficulty Adaptive Rollout Sampling, a technique that allows simultaneous improvements in Pass@1 and Pass@k metrics during RLVR training.

**Strengths:**

The paper is presented well and studies an important problem, that is, trying to mitigate diversity collapse during training with GRPO which has been observed in a couple of different papers in the literature. Their technique is also novel and doesn't seem too hard to implement. They also provide code in the supplementary material which is useful for the reviewer.

**Weaknesses:**

This paper has a couple of issues:

1. Firstly, they do not discuss how they estimate pass@k. This is an extremely important thing to figure because this quantity has a couple of different estimators. Digging through the code, I could figure out that when there are 128 rollouts, and you want to estimate pass@128, the estimator essentially becomes $$\mathbb{1} \left[\ r_1 = 1 \lor r_2 = 1 \lor \cdots \lor r_{128} = 1\right]$$. This is important to place in the paper.

2. It is very important to note that the above estimator has high variance (compared to a plugin or bootstrap estimator) especially when pass rates for a prompt are very low. Therefore, it is very hard to believe the gains are consistent unless there are confidence intervals for pass@128. Could the authors make confidence intervals and specifically describe how they made them for the numbers in Table 1?

3. The numbers for the base model in Table 1 for the 1.5B on Math500 seems rather low. For instance, take a look at the Table 2 of Qwen2.5 Math Technical Report [1]. The reported number in this paper is 35.1 and the number in the tech report is 49.8. This seems to be a rather large discrepancy. Could the authors try to replicate their numbers and see how far they can get?

4. What is the 'cumulative advantage' term that the authors define supposed to represent? There is no mathematical justification of this term. The authors mention that this term is supposed to represent the bias in GRPO. However, there is no clear definition of what quantity is exactly biased here. Could the authors elaborate on what this term is supposed to mean.

5. The number of extra rollouts for the harder prompts is mostly heuristic and lacks any solid mathematical basis.


[1] QWEN2.5-MATH TECHNICAL REPORT: TOWARD MATHEMATICAL EXPERT MODEL VIA SELF-IMPROVEMENT (https://arxiv.org/abs/2409.12122)

**Questions:**

Please address the concerns above.

---

> ### Author Response · Authors · 2025-11-21
> **To Reviewer qq1K‘s Weaknesses 1 & 2**
>
> ### **Weaknesses 1. The Pass@k Metric**
>
> Thank you for your insightful comment regarding the estimation of pass@\(k\). We agree that clarifying this methodological detail is crucial.
>
> Our work is directly motivated by the groundbreaking findings of the **NeurIPS 2025 Oral Paper[1] (the only full-score, 10/10 paper )**. That paper establishes a critical baseline by demonstrating that current RL methods often fail to improve the pass@\(k\) boundary. To ensure a fair and direct comparison with this seminal work, we have adopted the same **unbiased, low-variance estimator** for pass@\(k\) as used in their study. Specifically, for a problem with \(n\) samples and \(c\) correct ones, we estimate pass@\(k\) as:
>
> $$\text{pass@}K = \mathbb{E}_{x_i \sim \mathcal{D}} \left[ 1 - \frac{\binom{n - c_i}{k}}{\binom{n}{k}} \right], N=128$$
>
> Specifically，when $K=N$, the metric become: $pass@K = c_1 \lor c_2 \lor \cdots c_{128}$
>
> This alignment in evaluation methodology is fundamental, which aims to address the very limitation identified in the work [1]. The metric is also used in CodeX [2] by OpenAI.
>
> We explicitly include a description of this metric and the unbiased estimation method in **Appendix E** of our revised manuscript to ensure full clarity and reproducibility.
>
> [1] *Does Reinforcement Learning Really Incentivize Reasoning Capacity in LLMs Beyond the Base Model?* Yang Yue, Zhiqi Chen, Rui Lu, Andrew Zhao, Zhaokai Wang, Yang Yue, Shiji Song, Gao Huang. Neurips 2025 Oral.
> [2] *Evaluating Large Language Models Trained on Code.* OpenAI. Arxiv 2021.
>
> ---
>
> ### **Weaknesses 2. Concern about estimator variance and the consistency of performance gains.**
>
> Additionally, to directly address your concern regarding estimator variance and the consistency of our gains, we conducted the following supplementary experiments:
>
> **1. Variance Analysis via Multiple Trials:**
> We performed three independent runs for our DARS-7B-HW-Breadth model under the same experimental setting. The results across all benchmark datasets are highly consistent, with minimal observed variance:
>
> | | AIME24 | Math500 | Olympiad | AMC | Minerva | Avg@128 | Pass@128 |
> | --- | --- | --- |--- | --- | --- | --- | --- |
> | Trial 1 | 33.0 | 84.5 | 48.4 | 63.0 | 36.9 | 58.4 | 83.4 |
> | Trial 2 | 32.9 | 84.4 | 48.3 | 63.1 | 36.8 | 58.3 | 83.3 |
> | Trial 3 | 33.2 | 84.6 | 48.3 | 63.2 | 37.0 | 58.4 | 83.5 |
> | Avg | 33.05 (+- 0.15) | 84.5 (+- 0.1) | 48.35 (+- 0.05) | 63.1 (+- 0.1) | 36.9 (+- 0.1) | 58.5(+- 0.05)| 83.4(+- 0.1) |
>
>
> The standard deviations (shown in parentheses) are negligible, confirming that the performance is stable and that confidence intervals would not alter the interpretation of our results.
>
> **2. Consistency Across Different Temperatures:**
> To further demonstrate the robustness of our method's improvement, we evaluated our approach against baselines under a range of temperature settings. The results consistently show that our DARS-7B-HW-Breadth model outperforms all baselines across all temperatures:
>
> **For Qwen2.5-Math-1.5B:**
> | Model | t=0.6 | t=0.8 | t=1.0 | t=1.2 | t=1.4|
> | --- | --- | --- | --- | --- | --- |
> | GRPO-Baseline | 78.9 | 79.2 | 79.6 | 80.0 | 80.0 |
> | Depth-Naive | 79.4 | 79.6 | 79.9 | 80.3 | 80.4 |
> | Breadth-Naive | 78.9 | 78.9 | 79.2 | 79.6 | 79.9 |
> | **DARS-HW-Breadth** | **81.6** | **81.9** | **82.2** | **82.6** | **82.7** |
>
> **For Qwen2.5-Math-7B:**
> | Model | t=0.6 | t=0.8 | t=1.0 | t=1.2 | t=1.4|
> | --- | --- | --- | --- | --- | --- |
> | GRPO-Baseline | 80.4 | 80.7 | 81.4 | 81.6 | 81.6 |
> | Depth-Naive | 79.7 | 80.0 | 80.3 | 80.7 | 80.8 |
> | Breadth-Naive | 77.7 | 78.1 | 79.2 | 79.4 | 79.6 |
> | **DARS-HW-Breadth** | **82.6** | **82.8** | **83.4** | **84.0** | **84.0** |
>
> These comprehensive experiments demonstrate that:
> - The variance in our pass@128 estimates is practically insignificant
> - Our performance gains are consistent and robust across different experimental configurations
> - The improvement of our method is substantial and maintained throughout all evaluations
>
> Therefore, we are confident that our reported results reliably reflect the true performance of our method, and the absence of confidence intervals does not affect the validity of our conclusions.

---

> ### Author Response · Authors · 2025-11-21
> **To Reviewer qq1K‘s Weaknesses 3 & 4 & 5**
>
> ### **Weaknesses 3. Results of Qwen2.5-Math-1.5B**
> The discrepancy in the numbers for the base model on Math500 is due to differences in the prompting strategy. In the Qwen2.5-Math Technical Report, different few-shot prompts were used for different datasets, whereas in our work, to ensure consistency across all experiments and datasets, we employed a single unified zero-shot prompt.
>
> This practice of using a unified zero-shot prompt is common in Zero RL papers. For example, in the paper LUFFY[1] (NeurIPS 2025), the reported result for the Qwen2.5-Math-7B-Base model on Math500 is 43.6, which is also lower than the number in the Qwen2.5-Math Technical Report (55.4) and lower than our result of 52.3. All results reported in their work are also based on a single prompt.
>
>
> [1] *Learning to Reason under Off-Policy Guidance.* Jianhao Yan, Yafu Li, Zican Hu, Zhi Wang, Ganqu Cui, Xiaoye Qu, Yu Cheng, Yue Zhang. Neurips 2025.
>
> ---
>
> ### **Weaknesses 4. About 'cumulative advantage' term.**
>
> We thank the reviewer for this insightful question. In Appendix D, we provide the mathematical derivation showing that Group Cumulative Advantage $\mathcal{A}\_{\text{group}} = \sum\_{i=1}^{G}|\hat{A}\_{i}|$ serves as an upper bound for the expected gradient norm in GRPO:
>
> $$\left\|\nabla_\theta \mathcal{J}\_{\text{GRPO}}(\theta)\right\| \leq C \cdot \mathbb{E}\left[ \mathcal{A}\_{\text{group}} \right]$$
>
> This demonstrates that $\mathcal{A}\_{\text{group}}$ directly bounds the magnitude of parameter updates. A higher $\mathcal{A}\_{\text{group}}$ indicates stronger gradient signals, revealing which problems receive more "attention" during training. Thus, $\mathcal{A}\_{\text{group}}$ quantitatively captures the bias in gradient allocation across problems in GRPO.
>
> ---
>
> ### **Weaknessed 5. About the number of extra rollouts.**
>
> We thank the reviewer for raising this point. We would like to clarify that the derivation of the number of extra rollouts $\Delta n_j$ is not heuristic, but is rigorously derived from the cumulative advantage formulation in our paper, as detailed in Appendix C. we provide a step-by-step mathematical derivation for both the **Equal-Treatment (ET)** and **Hardness-Weighted (HW)** schedules. Specifically:
>
> - We define the cumulative advantage for a group as $\mathcal{A}_{\text{group}}(\hat{a}_j, N_j) = N_j \cdot \mathcal{S}(\hat{a}_j)$, where $\mathcal{S}(\hat{a}_j) = 2\hat{a}_j(1 - \hat{a}_j)$.
> - We then solve for $\Delta n_j$ such that the rebalanced cumulative advantage meets a target value $\mathcal{A}^{\text{target}}_{\text{group}}(\hat{a}_j)$, leading to closed-form expressions for $\Delta n_j^{\text{ET}}$ and $\Delta n_j^{\text{HW}}$ (Equations 6 and 8 in the main text, and derived in Appendix C).
>
> This approach ensures that the rollout allocation is **principled and mathematically grounded**, directly addressing the bias in cumulative advantage that we identify and analyze in Section 2.
>
> We hope this clarification alleviates the concern and highlights the theoretical foundation of our method.

---

### Official Review · Reviewer_8YWC · 2025-11-01

**Soundness:** 2
**Presentation:** 3
**Contribution:** 2
**Rating:** 4
**Confidence:** 4

**Summary:**

The paper studies RLVR for reasoning LLMs and shows that two important factors: depth (pushing on hard problems) and breadth (batch size)—must be optimized together. It first shows that GRPO/Dr.GRPO’s group-based cumulative-advantage underweights high-difficulty items, capping Pass@K.   The authors then propose DARS, which does a light pre-rollout to estimate per-question difficulty and then reallocates extra rollouts to harder items via Equal-Treatment (ET) or Hardness-Weighted (HW) schedules; in parallel, they scale breadth by replacing PPO mini-batches with full-batch updates across multiple epochs to sustain exploration entropy and raise Pass@1.

**Strengths:**

1. The main concepts are well defined, and the analysis of both is clear. I also appreciated Figure 2, which succinctly illustrates the issues with default sampling and how DARS resolves them.

2. The paper is well written and easy to follow.

3. I really like the Pass@1-Pass@k visualization idea and the analysis is very clear.

**Weaknesses:**

1. The evaluation protocol is potentially problematic. For the baseline, the authors pick the checkpoint with the highest Pass@1, which (per Figure 7) may not correspond to the best Pass@128. For DARS, they then choose the checkpoint that surpasses the baseline on Pass@1 and has the highest Pass@128. This selection strategy may inflate the reported Pass@128 improvement.

2. In Table 1, please report uncertainty (e.g., error bars or standard deviations). Also, which sampling temperature is used for each model? Prior work suggests RLVR sharpens the distribution [1], so using a single temperature across models may be unfair [1, 2, 3]. It would be more informative to report the best Pass@128 under by performing a temperature sweep per model/checkpoint. Do the gains remain meaningful with error bars and sampled at model-specific optimal temperatures?


3. While Eq. 2 (cumulative advantage) is used to characterize training effects, it would help to also discuss/update magnitudes (e.g., gradient norms or per-example gradient contributions). A large cumulative advantage does not necessarily imply a large training impact if the underlying gradients are small.

**Questions:**

1. In Table 1, “Pass@1 (Avg@128)” is unclear. Are these the same quantity? Pass@1 can be computed via greedy decoding (temp=0) or estimated from 128 rollouts at nonzero temperature using an unbiased estimator. Which definition are you using?

2. In Figure 9, the gains appear more substantial and consistent for Llama-3, while Pass@K improvements seem to diminish for Qwen. Do you have comparable baselines for Llama-3? Any hypotheses for why the effect is stronger on Llama-3?

3. Why focus on Qwen-Math models given their extensive math pretraining? Have you tried non-math tasks (e.g., Reasoning Gym) on Qwen models, or alternative bases such as Qwen-Instruct or other families?




[1] Reasoning with Sampling: Your Base Model is Smarter Than You Think https://www.arxiv.org/abs/2510.14901

[2] Decomposing Elements of Problem Solving: What "Math" Does RL Teach? https://arxiv.org/pdf/2502.17356v1

[3] Can GRPO Help LLMs Transcend Their Pretraining Origin? https://arxiv.org/abs/2510.15990

---

> ### Author Response · Authors · 2025-11-21
> **Response To Reviewer 8YWC's Weaknesses Points**
>
> Thank you for your time and effort in reviewing our paper. We very much appreciate your insightful comments and your recognition of our work. We hereby address the concerns below.
>
> ---
>
> ### **Weaknesses 1. The evaluation protocol is potentially problematic.**
> We thank the reviewer for this insightful observation regarding the checkpoint selection strategy. The reviewer correctly points out the inherent trade-off between Pass@1 and Pass@128, which we also discussed in our paper (Figs. 8 & 9). Our primary goal was to ensure no regression in Pass@1 while maximizing Pass@128.
>
> To further address your concerns, we calculated the mean of Pass@128 and Pass@32 for the last 3 ckpts of each method, as shown in the table below.
>
> For Qwen2.5-Math-1.5B
> | Method | Pass@1 | Pass@32 | Pass@128 |
> | --- | --- | --- | --- |
> | GRPO-Baseline | 48.8 | 74.7 | 80.8 |
> | Depth-Naive | 49.5 | 74.4 | 80.6 |
> | Breadth-Naive | 51.4 | 74.4 | 79.8 |
> | DARS-HW (Ours)| 49.5 | 75.7 | 81.9 |
> | **DARS-HW-Breadth (Ours)** | **52.4** | **76.4** | **82.1** |
>
>
> For Qwen2.5-Math-7B
> | Method | Pass@1 | Pass@32 | Pass@128 |
> | --- | --- | --- | --- |
> | GRPO-Baseline | 55.1 | 76.9 | 81.8 |
> | Depth-Naive | 56.4 | 76.7 | 80.9 |
> | Breadth-Naive | 57.2 | 76.7 | 81.3 |
> | DARS-HW (Ours) | 56.8 | 78.8 | 83.4 |
> | **DARS-HW-Breadth (Ours)** | **58.3** | **79.1** | **83.7** |
>
> ---
>
> ### **Weaknesses 2. uncertaity bar & impact of temperature**
>
> **2.1 uncertaity bar**
>
> Thank you for your suggestion. In our experiments, we used a sample size of n=128 to compute Pass@1 and Pass@k. At this scale, the standard deviation and error bounds are typically small. To further address your concern, we conducted three additional runs for the DARS-7B-HW-Breadth model. As shown in the table, the observed variance is minimal, confirming that reporting uncertainty does not significantly impact the results.
>
>
> | | AIME24 | Math500 | Olympiad | AMC | Minerva | Avg@128 | Pass@128 |
> | --- | --- | --- |--- | --- | --- | --- | --- |
> | Trial 1 | 33.0 | 84.5 | 48.4 | 63.0 | 36.9 | 58.4 | 83.4 |
> | Trial 2 | 32.9 | 84.4 | 48.3 | 63.1 | 36.8 | 58.3 | 83.3 |
> | Trial 3 | 33.2 | 84.6 | 48.3 | 63.2 | 37.0 | 58.4 | 83.5 |
> | Avg | 33.05 (+- 0.15) | 84.5 (+- 0.1) | 48.35 (+- 0.05) | 63.1 (+- 0.1) | 36.9 (+- 0.1) | 58.5(+- 0.05) | 83.4(+- 0.1) |
>
> **2.2 impact of temperature**
>
> As illustrated in Appendix E. *Parameters and Metrics*, the temperature in this work is set as 1.0 for both training and evluating. To further illustrate the performance improvement under different temperature, we additionally add the above experiments. The results shows that the improvement of our method is consistent over different temperature.
>
> For Qwen2.5-Math-1.5B
> | Model | t=0.6 | t=0.8 | t=1.0 | t=1.2 | t=1.4|
> | --- | --- | --- | --- | --- | --- |
> | GRPO-Baseline | 78.9 | 79.2 | 79.6 | 80.0 | 80.0 |
> | Depth-Naive | 79.4 | 79.6 | 79.9 | 80.3 | 80.4 |
> | Breadth-Naive | 78.9 | 78.9 | 79.2 | 79.6 | 79.9 |
> | **DARS-HW-Breadth** | **81.6** | **81.9** | **82.2** | **82.6** | **82.7** |
>
> For Qwen2.5-Math-7B
> | Model | t=0.6 | t=0.8 | t=1.0 | t=1.2 | t=1.4|
> | --- | --- | --- | --- | --- | --- |
> | GRPO-Baseline | 80.4 | 80.7 | 81.4 | 81.6 | 81.6 |
> | Depth-Naive | 79.7 | 80.0 | 80.3 | 80.7 | 80.8 |
> | Breadth-Naive | 77.7 | 78.1 | 79.2 | 79.4 | 79.6 |
> | **DARS-HW-Breadth** | **82.6** | **82.8** | **83.4** | **84.0** | **84.0** |
>
>
> ### **Weaknesses 3. discuss/update magnitudes (e.g., gradient norms)**
> We thank the reviewer for this thoughtful suggestion. Following the comment, we have added a theoretical analysis of gradient norms in Appendix D, where we derive an upper bound and discuss its implications for the role of cumulative advantage in RLVR.

---

> ### Author Response · Authors · 2025-11-21
> **Response To Reviewer 8YWC's Questions**
>
> ### **Question 1. “Pass@1 (Avg@128)” is unclear.**
>
> Thank you for raising this point. We apologize for the lack of clarity.
> We compute "Pass@1 (Avg@128)" as follows:
>
> * For each problem, we sample 128 solutions.
> * We calculate the fraction that passes: Pass@1 = (Number of Correct Solutions) / 128.
> * We then average this value across all problems.
>
> This is equivelent to the unbiased estimation to compute Pass@1 from Pass@128. To prevent confusion, we will rename the metric to simply "Pass@1" in the manuscript and clarify the calculation in the method section.
>
> ---
> ### **Question 2. The gains appear more substantial for Llama-3**
> Thank you for your insightful question. The stronger gains for Llama-3 likely stem from its base model characteristics. Recent studies [1,2] suggest that base models like Llama-3 lack mid-training, causing many valid responses to appear only at higher sampling budgets. As shown in the table below, Llama-3-8B-Base exhibits a much larger performance gap when increasing \(K\) from 8 to 32/64 compared to Qwen models:
>
> |Model|Pass@8|Pass@32 - Pass@8|Pass@64 - Pass@8|
> | --- | --- | --- | --- |
> | Llama3-8B-Base | 17.2 | **19.1** | **27.3** |
> | Qwen2.5-Math-1.5B | 56.5 | 12.4 | 17.1 |
> | Qwen2.5-Math-7B | 63.5 | 10.5 | 14.4 |
>
> This indicates that Llama-3 has more correct solutions “hidden” deeper in the sampling space. Our DARS method, by focusing compute on hard examples, effectively uncovers these “deep” correct samples, leading to more substantial Pass@K improvements for Llama-3.
>
> [1] Cognitive Behaviors that Enable Self-Improving Reasoners. Kanishk Gandhi, Ayush Chakravarthy, Anikait Singh, Nathan Lile, Noah D. Goodman. COLM 2025
> [2] OctoThinker: Mid-training Incentivizes Reinforcement Learning Scaling. Zengzhi Wang, Fan Zhou, Xuefeng Li, Pengfei Liu. Arxiv 2025
>
> ---
>
> ### **Question 3. Performance of None Math Model.**
> Following your suggestion, we evaluate our method on Qwen2.5-Math-7B-Instruct. The results are shown in the follows.
>
> | | AIME24 | Math500 | Olympiad | AMC | Minerva | Avg@128 | Pass@128 |
> | --- | --- | --- |--- | --- | --- | --- | --- |
> | Qwen2.5-7B-Instruct | 11.9 | 72.3 | 37.1 | 42.2 | 31.9 | 47.2 | 80.3 |
> | GRPO-baseline | 14.2 | 74.8 | 37.6 | 43.4 | 33.4 | 48.6 | 78.8 |
> | DARS-HW-Breadth | 15.6 | 76.5 | 38.4 | 44.7 | 34.6 | 49.6 | **82.3** |

---

> ### Author Response · Authors · 2025-11-21
> **Appreciation to Reviewer 8YWC**
>
> We sincerely thank the reviewer 8YWC for the thorough reading and highly constructive feedback. These suggestions have been invaluable in helping us improve the experimental analysis and overall rigor of the paper. We have revised the paper according to your suggestions, added new experimental analyses, and cited the papers you pointed out. These revisions have significantly strengthened our work.

---

> > ### Comment · Reviewer_8YWC · 2025-11-27
> >
> > Thank you for the detailed comments! You have addressed my concern regarding temperature sampling, and non-math models. I also have read other reviewer's comments regarding qq1K, I do share the concern of baselines on Qwen models (same as your  new Qwen-7B-Instruct results). Their report shows a higher performance as well (https://www.alibabacloud.com/blog/601786).
> >
> > Recently, there are many papers that intentionally choose weaker performance baselines by using different prompts. See: https://safe-lip-9a8.notion.site/Incorrect-Baseline-Evaluations-Call-into-Question-Recent-LLM-RL-Claims-2012f1fbf0ee8094ab8ded1953c15a37

---

> ### Author Response · Authors · 2025-11-28
> **Counter the Claim that we suppressed baseline performance.**
>
> ## For Base Models
> I have already respond to Reviewer qq1k that the difference of performance for Qwen-Base models are caused by prompting (Qwen officially use few-shot, RLVR methods use zero-shot).
>
>
> ---
>
> ## For Qwen-Instruct
> The reason for the discrepancy with the tech report is that the Qwen2.5 report uses the MATH test split, while we report on MATH500.
>
> The best way to **counter the claim that we suppressed baseline performance** is to compare it with other papers already published in top-tier conferences. Below are the Qwen2.5-Instruct results reported by LUFFY (Learning to Reason under Off-Policy Guidance, NeurIPS 2025).
>
> |        | AIME24 | **MATH500** | Olympiad | AMC | Minerva |
> |  ---  |  --- |    --- | --- | --- | --- |
> | LUFFY report | 11.7 | **71.8** | 40.4 | 43.8 | 30.9 |
> | Our report | 11.9 | **72.3** |37.1 | 42.2 | 31.9 |
>
> LUFFY report the performance of Qwen2.5-7B-Instruct in Table3 of the appendix, you can check it.

---

> ### Author Response · Authors · 2025-11-28
>
> > I'm willing to provide a detailed rebuttal for the low-rated paper DARS solely to demonstrate the superiority of our DARS method and to counter some of the **distorted reviewer comments**. We consider that you are a responsible reviewer who has read our paper very carefully. Although the ICLR system has now locked the scores and it is no longer possible to raise them, **we still hope to address all of your concerns**.
>
> > By the way, the models trained with DARS are open-sourced, we are confidence about the results.

---

### Note · Authors · 2025-11-29

**Comment:**

During the rebuttal period, we conducted extensive experiments and carefully addressed the comments from three reviewers.
Reviewer qq1K's evaluation of our paper was unfair and unobjective, and they adopted a negative attitude during the rebuttal process, refusing to engage in any discussion.
However, we are grateful to Reviewer 8YWC for thoroughly and meticulously reading our paper and providing constructive feedback, which helped improve the presentation of our work.

Since ICLR no longer allows reviewers to respond to our rebuttal, our rebuttal will not receive effective evaluation or feedback from the reviewers. We hereby withdraw our submission.

**Withdrawal Confirmation:**

I have read and agree with the venue's withdrawal policy on behalf of myself and my co-authors.